# A Microphysics Guide to Cirrus – Part 3: Occurrence patterns of cloud particles

Martine Krämer<sup>1,2</sup>, Nicole Spelten<sup>1</sup>, Christian Rolf<sup>1</sup>, and Reinhold Spang<sup>1</sup>

Correspondence to: Martina Krämer (m.kraemer@fz-juelich.de), ORCID ID: 0000-0002-2888-1722

**Abstract.** Cloud particle size distributions (PSDs) are crucial in determining the clouds physical and optical properties and hence their radiative feedback to the climate. Here we present unprecedented occurrence patterns of cloud particles derived from 270 hours of cloud measurements ( $\approx 975.000$  PSDs). The focus of the analysis is on cirrus clouds, but liquid and mixed-phase clouds are also shown. In particular, cirrus PSDs for cold to warm cirrus temperatures and microphysically thin to thick cirrus clouds are provided in a novel presentation as heat maps. The observations are accompanied by simulations of ice crystal growth in in situ-origin cirrus, showing that the maximum size to which the cirrus ice crystals can grow increases from approx. 60μm@T<200K to 230μm@T>220K. Crystals larger than this size are most likely of liquidorigin. The combined evaluation of observations and simulations allows the attribution of processes shaping the PSDs. Important results are that, with increasing temperature and cirrus thickness, the most frequent ice particles change from smaller and fewer crystals of in situ-origin to larger and more crystals of both in situ and liquid-origin i.e. the cirrus type changes from in situ- to liquid-origin. In addition, three characteristic ice crystal size ranges are identified. The nucleation / evaporation size interval ( $\sim$ 3–20 $\mu$ m), most frequent in the coldest, thinnest in situ-origin cirrus; the -most common- overlap size interval ( $\sim 20-230\mu m$ ), where both in situ-origin liquid-origin cirrus occur and the uplift / sedimenation size interval ( $>\sim 230 \mu m$ ), which consists mostly of liquid-origin ice crystals.

<sup>&</sup>lt;sup>1</sup>Institute of Climate and Energy Systems (ICE-4), Research Center Jülich, Jülich, Germany

<sup>&</sup>lt;sup>2</sup>Institute for Physics of the Atmosphere (IPA), Johannes Gutenberg University, Mainz, Germany

#### 1 Introduction

30

45

50

In Parts 1 and 2 of the study (Krämer et al., 2016; Krämer et al., 2020, hereafter referred to Cirrus Guide I and II), first a detailed guide to the formation and evolution of cirrus cloud is provided, compiled from extensive model simulations combined with field observations, followed by the presentation of climatologies of clouds and humidities from comprehensive in situ as well as remote sensing observations. Both studies were motivated by the continuing lack of understanding of the microphysical and radiative properties of cirrus clouds, which was and still is one of the greatest uncertainties in predicting the Earth's climate (IPCC, 2014).

Wang et al. (2020) pointed out that the climate is sensitive to the size of cloud ice particles and that reducing this uncertainty would reduce the mean state uncertainty in climate simulations. Therefore, knowledge of cloud particle size distributions (PSDs) under different atmospheric conditions is important. However, measurements of PSDs are difficult to accomplish, because clouds are always located at a certain altitude in the atmosphere. In addition, the entire cloud particle size range of ~3 to more than 1000 µm cannot be covered with one instrument and also, an almost undisturbed sampling cloud particles across their entire size range has only been successful for about 15 years.

Larger data sets of ice cloud PSDs has so far been compiled by Field et al. (2005); Heymsfield et al. (2013); Jackson et al. (2015); Lawson et al. (2019) and Bartolomé García et al. (2024). Field et al. (2005) and Heymsfield et al. (2013) derived unimodal ice cloud gamma size distributions based on the observational PSD data sets, while Jackson et al. (2015) and Bartolomé García et al. (2024) considered multiple or bimodal modes in PSD gamma functions.

The data sets on which the studies are based are very different. •Field et al. (2005) measured stratiform ice clouds around the British Isles at temperatures between -3 and -50°C (270 – 223 K) in the size range 25 to 6400 μm during 16 flights. •Heymsfield et al. (2013) collected data from 10 field campaigns from the Arctic to the tropics, covering 800.000 km of cloud horizontal pathlengths. Convective and stratiform clouds are sampled between 0 and -86°C (273 – 187 K) in the size range 100 – 3000 μm for the observations above -60°C. Between -60 and 86°C, the size range between 5 – 1000 μm is analyzed. Average size distributions in 10°C increments are also presented in this study. Sub-200-μm-diameter particles dominate the in situ generated cirrus between -60 and -86°C. •Jackson et al. (2015) present PSDs from the Sparticus field campaign in mid-latitude ice clouds over the U.S. Central Plains. Synoptic and convective ice clouds are presented for the temperature range -20 to -70°C (253 – 223 K) and particle sizes from 15 to 3000 μm diameter during 101 missions and ~25 hours of cloud observations. Heat maps of synoptic and convective ice cloud PSDs are shown in the study for different temperature intervals. •Lawson et al. (2019) present a review of 22 airborne field campaigns, characterizing ice cloud particle shapes and PSDs in convective anvils and in situ formed cirrus between ~2 - 3000 μm in the temperature range 0 to -90°C. The study

focuses on characterizing the shapes of ice particles in relation to their size under various atmospheric conditions.

•Bartolomé García et al. (2024) presents the PSDs contained in the JÜLich In situ Airborne database (JULIA) database (part of the database presented in the Cirrus Guide II by Krämer et al., 2020), one of the largest PSD collection, consisting of measurements from 11 field campaigns in the temperature range 27 to -88°C (300-185 K) and particle sizes from 3 to 1000 µm diameter. 163 flights with a total of 270.4 hours (≈ 975000 PSDs) in cirrus, mixed phase and liquid clouds are combined (for more detail see Section 3). The present study is based on this dataset, but focuses on the processes that shape cirrus PSDs. while Bartolomé García et al. (2024) derived PSD parameterizations. This extensive database, covering the entire temperature range and nearly the entire range of cloud particle sizes and concentrations is excellently suited to explore the appearance of different clouds. In particular, we are investigating the relationship between environmental conditions and the occurrence of cloud particle sizes in the PSDs to identify the most frequent size ranges in different clouds. This analysis will also lead to a better understanding of the processes that shape the PSDs. Although the focus here is on cirrus clouds, an overview of mixedphase and liquid clouds is also provided. To support the analyses of the cirrus PSDs, we also perform simulations of the maximum possible size to which in situ formed ice crystals can grow under different conditions. The scientific approach of the study is described in Section 2, Section 3 explains the database and further processing of the PSDs. The overview on mixed-phase and liquid clouds is shown in Section 4. In Section 2.3.3 and Section 5, the maximum size of in situ ice crystals and the occurence pattern of cirrus cloud particles are presented. Section 6 summarizes and concludes the study.

## 2 Methodology

80

55

To explain the appearance of certain particle sizes in cloud PSDs, first of all a suitable representation of the measurements is important. Note that in order to make use of the entire database, the PSDs from different cloud spectrometers are synchronized to a logarithmically equidistant grid as described in Section 3.2. Using two research flights as examples, we demonstrate how to best represent cirrus ice particle occurences in PSD measurements (Section 2.1). However, the basis for interpreting the sizes occurring in PSDs is an understanding of the processes that shape the size distributions, which are briefly introduced in Section 2.2. Another important component for this interpretation are the sizes to which ice crystals formed in situ can grow under different environmental conditions. These sizes are determined with the help of simulations, which are described in Section 2.3.

#### 2.1 Presentation of PSDs





Whether some of the physical processes that form the PSDs can be identified from the measurements depends to a certain degree on how these are displayed. Therefore, we would like to take a closer look at the presentation of the PSDs in this section.

Cloud PSDs are usually displayed as averages over a number of observations, as shown as an example in Figure 1 (panels a and c) for two flights through quite different cirrus clouds during the StratoClim campaign in 2017 out of Nepal (see also Section 3.1). This presentation implies that cloud particles are always present across the entire size range. Panels b and d of Figure 1 shows the PSD time series of the two flights along the aircraft flight paths. The upper boxes in (b) and (d) show temperature (red), pressure (green) and relative humidities with respect to ice (RH<sub>ice</sub>, turquoise) and water (RH<sub>w</sub>, blue). In the lower boxes, particle sizes between 0.6 to  $1000 \, \mu m$  diameter are shown, color coded by their normalized concentration (dN/dlogD<sub>p</sub>). From this portrayal of the PSDs measured every second it is obvious, that ice particles are not always present at T<-38°C. During the flight on 8 August (left panel), ice particles between 3 and 20  $\, \mu m$  are often very sparse or not even present and the maximum ice crystal sizes greatly differ. This can also be seen in the flight on August 10th (right panel), but here also high concentrations of small ice particles are measured and, also the size range between 3 and 20  $\, \mu m$  is more often occupied. All of the information that can be seen from the time series of the PSDs is not reflected in the display as an average.

A more informative, but still quite condensed representation of the PSDs is provided by so-called *PSD* heat maps. In this portrayal, the two flights are shown in panels a and b in Figure 2. The mean PSD can still be seen as a black line, the median PSD is in green. The color code shows the frequencies of ice crystal concentrations across the size range, with the black and white contours enclosing the size range covering 90 and 50% of the data, respectively.

Another informative portrayal of cirrus cloud particle sizes and concentrations are  $N_i - \overline{R}_i$  heat maps (see Figure 2, panels b and d), i.e. heat maps in the particle number—mean radius space, with  $N_{ice}$ : total ice crystal number and  $\overline{R}_{ice}$ : mean size of ice crystals  $^1$ . The black lines represent constant ice water contents (IWCs), the region containing 90% and 50% of all data points is also enclosed by the black and white contour lines. This illustration does not include details of the ice crystal sizes, but instead includes the ice water content so it is a condensed representation of the microphysical properties of the clouds.

How to interpret the two cirrus clouds shown in Figure 2 in the two ways of representation is discussed in the next section, after the processes that shape the PSDs have been explained.

 $<sup>^{1}\</sup>overline{R}_{ice}$  is the mean mass ice crystal radius;  $\overline{R}_{ice} = \left(\frac{3 \cdot IWC}{4\pi \rho \cdot N_{ice}}\right)^{1/3}$  with  $\rho$ = 0.92 g/cm<sup>3</sup>.

#### 2.2 Processes shaping PSDs





There are several reasons why different size ranges can appear in cirrus PSDs. On the one hand, cirrus clouds can have two different origins - in situ-origin or liquid-origin - with very different cloud particle sizes, and on the other hand, the cloud particles change their size with the ambient conditions over the course of their lifetime.

Cirrus of in situ-origin form at temperatures below about -38°C (235 K) heterogeneously or homogeneously on ice nucleating particles containing an insoluble impurity or supercooled solution aerosol particles (Luebke et al., 2016; Krämer et al., 2016). The ice particle sizes evolve over time. First, small ice particles appear in the size range 1-10 µm (see Figure 1, right panel, the reddish colors mark high ice particle numbers nucleating in a fresh homogeneous ice nucleation event). Depending on the environmental conditions such as cooling/warming rate ( $\hat{=}$  negative/positive vertical velocity = updraft/downdraft), temperature and resulting super- or subsaturation, these particles grow or shrink by diffusional growth/sublimation at a faster rate as the cooling or warming rate increases (Jensen et al., 2024). Matured ice clouds have, given that no new ice nucleation event occurs, ice particle sizes in the range between tens and hundreds of microns, whereby the larger ice particles seddle out (see Figure 1, panel c, shortly after 6:00 UTC). In case a new ice nucleation event occurs during the lifetime of a cirrus, the PSDs of the events overlap. In the dissipation phase of the cirrus, when the cirrus is in subsaturated conditions, the PSDs shrink back in size, dependent on the warming rate ( $\hat{=}$  downdraft), temperature and resulting subsaturation.

Liquid-origin cirrus consist mostly of ice crystals larger then  $\sim 50 \mu m$  (Costa et al., 2017); they form as liquid clouds from farther below containing droplets of sizes  $< 50 \mu m$ , which freeze heterogeneously and grow in slower updrafts to large sizes (often  $> 500 \mu m$ , much larger than the sizes achieved in in situ-origin cirrus) due to the Wegener-Bergeron-Findeisen process. In fast updrafts like in convective clouds, supercooled liquid droplets, which are still present at around 235 K (-38°C), freeze homogeneously. In panel d of Figure 1, a liquid-origin cirrus overlapping an in situ-origin is marked. This mixture of the two cirrus types happens often, since liquid-origin cirrus are lifted in a certain updraft to altitudes where in situ-origin formation can happen. In case the updraft is strong enough,  $RH_{ice}$  can reach the homogeneous freezing threshold and a new ice nucleation event occurs.

With the background knowledge of the processes shaping cirrus cloud PSD, we discuss now the PSD and  $N_i - \overline{R}_i$  heat maps of the two exemplary PSDs shown in panels a, c and b, d in Figure 2. From the PSD heat map of the flight at 0808, it can be seen that the ice crystals <20  $\mu$ m do not occur frequently, the core size range of the PSD is between 20 and 100  $\mu$ m and around 0.05cm<sup>-3</sup> in concentration. During flight on 1008, the ice crystals <20  $\mu$ m occur much more frequently, the core size range is between 3 and 100  $\mu$ m and the concentration between 1 and 0.01 cm<sup>-3</sup>. Ice crystals >300  $\mu$ m are not very common in both flights.

From this illustration it can be seen that the cirrus clouds on flight 0808 were mature, as there were very few small ice crystals present, and that most likely in situ-origin cirrus clouds predominate, as the large ice crystals were also not very frequent. Flight 1008 looks similar, except that there was strong nucleation of small ice particles, probably caused by homogeneous freezing in strong convective conditions. This means that the main structures visible in the time series in Figure 1 can be found in this representation.

From the portrayal in the  $N_i - \overline{R}_i$  space it is also visible that flight 0808 had fewer and slightly larger ice particles than flight 1008. In addition, it can be seen that the main area of the cirrus is below the 10 ppmv IWC line in both flights, which indicates in situ-origin cirrus (Li et al., 2023).

# 155 2.3 Simulations of in situ-origin ice particle sizes




Ice particles in cirrus clouds of liquid origin can be much larger than those of in situ origin because the water vapor content further down in the atmosphere, where the liquid-origin cirrus form is much higher than the water vapor available at the cold conditions of in situ-origin cirrus. To get an impression on its dependence on temperature: the amount of water vapor at water saturation,  $H_2O_{\rm sat,ice}$ , is 3, 35, 300 ppmv at 190, 210, 230 K (see Table 1).

To better distinguish between in situ-origin and liquid-origin cirrus, it is therefore helpful to know how large in situ ice crystals can grow, because if larger ones are present in the measured PSDs, they are most likely of liquid-origin. To this end, we have derived maximum sizes that in situ ice crystals can reach for three different temperature ranges from a climatology of in situ-origin cirrus simulated with the detailed microphysical ice cloud model MAID and presented in the Cirrus Guide I (Krämer et al., 2016). MAID includes Lagrangian ice particle tracking, which allows to follow the size evolution of individual ice particles.

# 2.3.1 MAID: Model for Aerosol and Ice Dynamics

MAID (Bunz et al., 2008; Rolf et al., 2012, for more detail see also the Cirrus Guide I) is a microphysical ice process box model with Lagrangian ice particle tracking. The ice nucleation processes are heterogeneous freezing after Kärcher et al. (2006) and homogeneous freezing after Koop et al. (2000). The heterogeneously freezing ice nucleating particles (INPs) can be varied in concentration as well as the freezig threshold (low: 110-120%(T), high: 130-150%(T)). Ice crystal sedimentation is treated in MAID following the sedimentation scheme of Spichtinger and Gierens (2009). MAID is a size bin resolving microphysical model with Lagrangian ice particle tracking, that means once the ice particles have formed, each particle develops individually in size by diffusional growth, sublimation and sedimentation. Consequently, using MAID it is not only possible to follow the evolution of bulk parameters IWC, N<sub>ice</sub> and the humidity, but to follow also the development of the ice crystal size spectrum.

# 2.3.2 In situ-origin cirrus climatology







The Cirrus Guide I climatology (Krämer et al., 2016) contains around thousand model runs covering nearly the entire atmospheric range of cirrus cloud conditions. In situ-origin cirrus clouds are simulated for temperatures varying between 180 and 235 K in 10 K steps and six different updrafts between 1 and 300 cm/s. The simulations, each starting at RH<sub>ice</sub> = 90%, are performed with constant large scale updrafts as well as by superimposing small scale temperature fluctuations. The heterogeneously freezing ice nucleating particles (INPs) are varied in concentrations between 0.001 and 1 cm<sup>-3</sup> in four steps and the freezing humidity threshold is assumed to be either low ( $\sim$ 110-120%) or high ( $\sim$ 130-150%) and increases with temperature. The homogeneously freezing aerosol particles are assumed to be supercooled binary solution particles with a concentration of 300 cm<sup>-3</sup> and a mode size of 200 nm. The scenarios of the cirrus climatology last until dynamical equilibrium is reached, i.e. the ice particles have reached their maximum possible size.

In Figure 3, an exemplary scenario of the development of in situ-origin cirrus is presented in the IWC-temperature space. Simulations are shown for various updrafts (see color code), starting at different temperatures. For clarity of the presentation we show a scenario with constant updrafts. Note that decreasing temperature corresponds to time evolution. In panel a of Figure 3, the color code corresponds to updraft, panel b is color coded by the ice crystal concentration  $N_{\rm ice}$  and panel c by the mean ice crystal size  $\overline{R}_{\rm ice}$ . The three panels show the relationship between updraft,  $N_{\rm ice}$  and  $\overline{R}_{\rm ice}$  and also IWC as a function of temperature. In weak updrafts (panel a, green),  $N_{\rm ice}$  is low (panel b, pale blue) and IWC decreases with temperature because of the decreasing amount of available water vapor to grow the ice crystals. Accordingly, the lower the temperature, the smaller the ice crystals (panel c, pale green). However, they are largest at these weak updrafts, in particular at the warmer temperatures (dark green). The stronger the updrafts (see color code in panel a), the more but smaller ice crystals are formed, because the water vapor distributes on more ice crystals. This means that the smallest (largest) ice crystals are found at the lowest (highest) temperatures and strongest (weakest) updrafts.

We have evaluated the scenarios of the simulated cirrus climatology formed by large-scale cooling superimposed with temperature fluctuations, because these are the most realistic conditions. From all simulations starting at 190, 210 and 230 K, those with the largest ice crystals are selected for each updraft. Based on these sizes, a maximum ice crystal diameter ( $D_{\rm ice}^{\rm max}$ ) is derived for each temperature range. The results are summarized in Table 1 and discussed in the next Section.

## 2.3.3 Maximum sizes of in situ-origin cirrus

The simulations that produced the largest ice crystals (diameter  $D_{\rm ice}^{\rm max}$ ) are listed in Table 1 for the five updrafts and three temperature ranges. These are the ones with the lowest number of efficient INPs (0.001)

cm<sup>-3</sup> = 1 L<sup>-1</sup>, freezing threshold  $\sim$ 110-120%). In addition, the corresponding N<sub>ice</sub> and IWC are shown for five different air parcel updrafts.

What can be seen from Table 1 is, as expected from the discussion of Figure 3 in the previous section, that the colder the temperature the thinner are the cirrus in terms of IWC and the smaller are the largest ice particles  $D_{\rm ice}^{\rm max}$ . At 190 K,  $D_{\rm ice}^{\rm max}$  is found to be between 5 to 56 µm, which is in good agreement with Jensen et al. (2010). They simulated the evolution of ice PSDs at 190 K in weak updrafts for a time period of  $\sim$ 3.5 hours and found maximum sizes of  $\sim$ 15µm in slight supersaturations. It is to note that at the coldest temperatures the largest possible ice crystals appear via heterogeneous ice nucleation (het) at low concentrations of INPs. Homogeneous freezing, that occur in all simulations after the initial heterogenous freezing event, produces ice crystals with smaller sizes, mostly due to a higher  $N_{\rm ice}$  on which the water vapor distributes.







Vice versa, the warmer the temperature and the weaker the updraft the larger is  $D_{\rm ice}^{\rm max}$ . Also, the IWC and the ice crystal concentration  $N_{\rm ice}$  are higher at warmer temperatures. Here, the largest crystals formed after the initial heterogeneous freezing in a subsequent, homogeneous ice nucleation event (het-hom) that produces more ice particles. They can grow to large sizes because of the amount of water at 230 K is hundred times larger than at 190 K. Note that due to the superimposed small scale temperature fluctuations, the vertical velocity at the point of ice nucleation deviates from the large-scale updraft. Therefore, the ice crystal concentration  $N_{\rm ice}$ , that would increase with increasing large-scale updraft in homogeneous freezing, varies with the respective prevailing small-scale updraft. However, it can be seen that  $N_{\rm ice}$  and IWC are correlated with each other, as shown by Krämer et al. (2020).

Remarkably, the largest simulated in situ formed ice particles have a size of only 224  $\mu$ m, which is considerably smaller than the largest cirrus ice particles from the observations (1000  $\mu$ m, see Figure 6, left column; there might be even larger ice particles present, but not recorded with the applied instrumentation). The maximum ice crystal sizes of ice from the MAID simulations are supported by observations of Wolf et al. (2018), who found an average maximum size of  $\sim$  140  $\mu$ m in their measurements in in situ-origin cirrus.

Aggregation of the in situ-origin ice crystals could produce larger crystals, but this process plays a -minor-role only at the warmest cirrus temperatures (Spichtinger, 2023; Sölch and Kärcher, 2010), because aggregation has a strong dependence on the ice particle size and weakens with ice mass, number concentration and temperature. Therefore, aggregation occurs mainly in fall streaks, i.e. at temperatures that are usually higher than those considered here, where the ice particles are large. This can also be seen in Gallagher et al. (2005), where aggregation is observed in fall streaks at temperatures between about 230 and 240 K, and is also discussed in a modelling study by Sölch and Kärcher (2010). Consequently, the larger ice particles in the observations can be considered to be of liquid-origin, confirming the results of the Cirrus Guides I and II.

Overall, a conservative estimate from the simulations is that in the three temperature ranges  $\sim$ 190, 210, and 230 K, ice crystals formed in situ do not grow larger than 60, 120, and 230 µm, respectively. However, we cannot rule out that at  $\gtrsim$  230 K, in situ-origin ice particles larger than  $D_{\rm ice}^{\rm max}$  may also occur, resulting from aggregation.

#### 3 Cloud PSD database

As mentioned in the introduction, the study uses data from the JULIA database (Bartolomé García et al., 2024), which should briefly be described here.

# 3.1 Field campaigns and instruments

Cloud PSDs were recorded every second during eleven field campaigns by using different pairs of cloud spectrometers, one to measure smaller cloud particles using light scattering techniques and the other to detect larger cloud particles by means of optical array probes. The instruments, their measurement methods, and the uncertainties are described in the first two parts of the study (Krämer et al., 2016; Krämer et al., 2020) and in Baumgardner et al. (2017); we therefore refer to these papers for further information.

- Seven campaigns with the NIXE-CAPS:
- 260 Coalesc 2011 (on board of the FAAM BAe-146);

Verdi 2012, Racepac 2014 (AWI Polar-5/6);

Acridicon 2014, ML-Cirrus 2014,

Cirrus-HL 2021 (HALO)

and Stratoclim 2017 (Geophysica);

- two instruments are integrated in the NIXE-CAPS (NIXE-Cloud and Aerosol Particle Spectrometer), the CAS-Depol (Cloud and Aerosol Spectrometer with depolarization, size range 0.6–50 μm) and the CIPgs (Cloud Imaging Probe with grey scale, size range 7.5-937.5 μm)
  - Two campaigns with the combination of CDP+2-DC:
- 270 Start\_08, Contrast 2014 (NCAR G5-HIAPER);
  - the CDP (Cloud Droplet Probe) covers a size range of 2–50  $\mu$ m, the 2D-C (Two-dimensional cloud probe) a size range of 60–1100  $\mu$ m; all particles are assumed to be from clouds.
  - Two campaigns with FCDP+2-DS:
- 275 Attrex 2014 (Global Hawk)

and Posidon 2016 (NASA WB-57);

the FCDP (Fast Cloud Droplet Probe) covers a size range of 1–50  $\mu$ m, the 2D-S (Two-dimensional stereo probe) a size range of 25–3005  $\mu$ m.

Particles larger than 3 μm are assumed to be cloud particles. The PSDs of the NIXE-CAPS and the FCDP+2-DS are merged between 20 and 25 μm, those of CDP+2D-C at 55 μm. The instruments and campaigns are described in more detail in Costa et al. (2017), Krämer et al. (2020) and Jurkat-Witschas and et al. (2024).

The measurements cover the temperature ranges of cirrus as well as mixed phase and liquid clouds. 285 Emphasis of this study are cirrus clouds. As additional parameters in cirrus clouds, IWC,  $N_{\rm ice}$  and  $\overline{R}_{\rm ice}$  are derived from the PSDs. To calculate the IWC, the m-D (mass dimension) relation shown in Krämer et al. (2016)  $^2$  is used.

As mentioned in the introduction, during 163 flights 137.2 hours were spend in cirrus ice clouds, 69.4 in mixed-phase and 63.8 in liquid clouds. An overview on the flights is given in Figure 4 (panel a), color coded by field campaign. Panels b and show the in-cloud temperatures with regard to latitude and altitude.

#### 3.2 PSD synchronization





The aim of the study is to analyse the cloud PSDs measured with the instruments described in the previous section. However, the gridding of the size intervals of the various instruments is different and irregular, so that the original measurements cannot be combined. For this reason, the irregular PSD size grids are synchronized to a logarithmically equidistant grid. Figure 5 shows examples of PSDs in 5K temparature intervals from the NIXE-CAPS instrument during the StratoClim field campaign, the FCDP+2D-S during ATTREX and the CDP+2D-C during CONTRAST are shown. The original size gridding is displayed in the left column, it is visible how irregular and different the size intervals of the instruments are. The PSDs based on the synchronized size grids are shown in the right column, demonstrating that the datasets can be combined now. An overview of the synchronized PSDs from each of the eleven field campaigns is shown in Appendix A, Figure A.1.

Finally, looking at the transitions between the PSDs of the instrument pairs reveals no discrepancies so large that the measurements would have to be discarded. The transitions falls within the known instrumental uncertainties, which unfortunately are generally not insignificant for cloud spectrometers.

After synchronizing the PSDs to the same size grid, data points can now be combined independently of the instruments and PSD and  $N_i - \overline{R}_i$  heat maps can be created. An overview of the PSD heat maps from each of the eleven field campaigns is given in Appendix A, Figure A.2,  $N_i - \overline{R}_i$  heat maps are presented in Figure A.3.

 $<sup>^{2}</sup>$ m = a · D $^{(b)}$  with a = 0.082740, b = 2.814 for D < 240 $\mu$ m and a = 0.001902, b = 1.802 for D > 240 $\mu$ m.

#### 4 Overview of cirrus, mixed-phase and liquid clouds






Cloud PSD heat maps of cirrus, mixed-phase and liquid clouds are displayed in variuos temperature ranges to provide an overview of the most common cloud particle sizes and concentrations contained in the PSD datasets. Figure 6 shows the heat maps for cirrus (a-c), mixed-phase (d-e) and liquid clouds (f) in their respective temperature ranges, namely cirrus clouds between 180-235 K (~ -90 to -38°C), mixed-phase clouds in the range 235-273 K (-38 to 0°C) and liquid clouds for temperatures >273 K (> 0°C). The mixed-phase and cirrus clouds are divided into sub-ranges, with the coldest temperature range representing measurements in the tropical tropopause layer (TTL, panel a).

Ice particle sizes in the TTL are found throughout the instrument's measurement range, with the most frequent sizes being  $<100~\mu m$ . What mainly happens in cirrus clouds as the temperature rises is that increasingly more larger ice crystals appear and therefore the shape of the median and mean PSDs (green and black lines) flatten. A more detailed dicussion on the development of the ice particle occurrence pattern in cirrus clouds is presented in Section 5.

Panels d-e of Figure 6 present mixed-phase clouds in the temperature ranges 235-255K and 255-275K. At the colder temperatures (panel d), the pattern of the most frequently appearing ice crystals differ not greatly from the next colder temperature interval in cirrus clouds (panel c). This is not surprising, because the warmest cirrus clouds contain a large portion of liquid-origin cirrus, which are glaciated mixed-phase clouds that are uplifted to higher altitudes. One difference, however, is that higher concentrations of smaller ice crystals are found in the mixed-phase clouds. This is because above 235K supercooled liquid droplets that have not yet frozen can still exist. However, their frequency is low, that means most of the clouds are completely glaciated due to the Wegener-Bergeron-Findeisen (WBF) process that transfers the many small liquid drops into few large ice crystals when the environment is subsaturated with respect to liquid water but supersaturated with respect to ice (see e.g. Costa et al., 2017). In the next warmer temperature interval (255-275K, panel e) the occurrence pattern changes drastically. Most common are now high concentrations of smaller cloud particles ( $\lesssim 30~\mu m$ ), while larger ones ( $\gtrsim 100~\mu m$ ) are much rarer. This indicates that at the warmer temperatures, the WBF process has not yet started and thus many small liquid drops coexist with few larger ice crystals.

Liquid clouds are displayed in Figure 6, panel f. The shape of the mean PSD is dominated by small cloud droplets, as for the coexistence mixed-phase clouds (panel e). Even fewer cloud particles larger than a few hundred micrometers are present; these are mostly ice particles that have sedimented down from above. On the other hand, more drizzle drops are present, which are cloud particles between 50 and  $\sim$ 300  $\mu$ m. These are the first droplets that freeze in mixed-phase clouds and grow into larger ice crystals. The most frequently observed particles are medium concentrations in the range of 3 to 10 micrometers. However, these are most likely not cloud droplets but aerosol particles in cloud free air, as the warm temperatures are found near the

ground, where larger aerosol particles are also present, albeit in lower concentrations than cloud droplets.

Our measurements did not separate cloud and aerosol particles, but Dollner et al. (2024) recently developed

an algorithm that makes this possible.

# 5 Occurence pattern of cirrus cloud particles


The cirrus cloud PSDs include 137.3 hours of measurements. Based on this large amount of data, we take a closer look at the microphysical structure of cirrus clouds as a function of temperature and IWC. In Figure 7, nine panels of cirrus PSDs are shown where the three rows represent (from top to bottom) the temperature ranges 180-200 K (coldest cirrus), 200-220 K (middle temperature cirrus) and 220-235 K (warmest cirrus) and the three columns the IWC ranges (from left to right) 0.001-0.2 ppmv (thin cirrus) , 0.2-20 ppmv (median thick cirrus) and 20-2000 ppmv (thick cirrus). For each temperature range, the bounding water vapor saturation mixing ratios ( $H_2O_{\rm sat,ice}$ ) are shown above the three tiles as an approximate amount of available gas phase water vapor.

The green and black lines represent the mean and median PSDs of the respective IWC-T tile. The vertical blue lines mark the maximum size of ice particles that can grow from the available water vapor through in situ ice formation (in situ D<sub>ice</sub><sup>max</sup>, see Section 2.3.3). The yellow-red color code indicate the frequencies of the ice particles across the entire size range. In this representation of PSDs, the average concentrations of different ice particle sizes is displayed in combination with their respective frequencies of occurrence, thus providing an additional important information about the ice particle population.

Accompanying Figure 7, characteristics of the PSDs, such as the maximum observed ice crystal sizes  $(D_{ice}^{max,obs})$  together with the maximum ice crystal sizes simulated for in situ-origin cirrus  $(D_{ice}^{max,sim})$ , as well as the observed core size ranges  $(D_{ice}^{core})$ : the size ranges enclosing 90 | 50 % of the observed data points), are summarised for the nine IWC-T tiles in Table 2. In addition, bulk cirrus properties (median  $N_{ice}$ ,  $\overline{R}_{ice}$  and  $RH_{ice}$ ) of the nine IWC-T tiles, derived from the climatologies presented in Krämer et al. (2020) (Figure 6, Cirrus Guide II) are displayed in Table 3. These numbers summarize key findings from the climatologies which will aid in understanding the differences in the PSDs. For more detail we recommend Krämer et al. (2016) and Krämer et al. (2020).

#### 5.1 The coldest cirrus (180-200K)

The coldest cirrus (180-200K) are shown in the upper row of Figure 7a-c. The altitude range of the observations is 14-19km, the data were collected in the TTL (see Figure 4).

In the thinnest cirrus (*IWC*: 0.001-0.2ppmv, Figure 7a), the median ice crystal concentration  $N_{\rm ice}$  is low (1.2 L<sup>-1</sup>, see Table 3) and the median RH<sub>ice</sub> is slightly supersaturated (106%). In the size range  $D_{\rm ice}$  below  $\sim$ 10µm, where nucleation of new ice particles takes place, the  $N_{\rm ice}$  (black curve) achieved are lowest

compared to the other IWC ranges in this temperature regime (Figure 7b, c); this is especially visible in the mean N<sub>ice</sub>, black curves). This suggests, in agreement with the findings of Krämer et al. (2016) and Krämer et al. (2020), that the larger IWCs are associated with higher updrafts, as the ice nucleation rate then increases. Krämer et al. (2016) and Krämer et al. (2020) show that low IWCs are tied to slow updrafts where only few ice particles are produced, either heterogenously or homogeneously. Heterogenous freezing usually results in few ice crystals due to the limited number of available INPs, and similarly only few ice crystals are nucleated by homogeneous freezing in slow updrafts.

Another characteristics of the coldest, thinnest cirrus PSD frequency distribution is that the overall ice particle size range is narrow, with a maximum size not exceeding  $\sim 350 \mu m$  in diameter (see also Table 2). However, it is notable that the largest sizes of the core PSD region containing 90% of the data points (black contour lines) is only  $\sim 70 \mu m$  and the region enclosing 50% of the data points (white contour line) is even only  $\sim 40 \mu m$ . Correspondingly, the median of the mean mass  $\overline{R}_{ice}$  is also small with  $7 \mu m$ .





The reason that no larger ice crystals are present in the coldest and thinnest cirrus is twofold: (i) First, the ice crystals, mostly of in situ-origin cannot grow to large sizes at these low temperatures. Due to the cold temperatures, the amount of available water to grow the ice particles is as small as 1-10 ppmv (as indicated by  $H_2O_{\rm sat,ice}$  in the headline above the first row of Figure 7). This interpretation is supported by the model simulations shown in Section 2.3. From Table 1 it is visible that at cold temperatures around 190 K, the largest ice particles can grow to an in situ  $D_{\rm ice}^{\rm max}$  of  $\sim\!60~\mu{\rm m}$  (also indicated in the headline of Figure 7 and as vertical line in the top three panels), which marks approximately the upper limit of the size range containing 90% of the data.

(ii) Second, larger liquid-origin ice crystals can either sediment from above or are uplifted from below. However, these cold temperatures correspond to altitudes so high (≥15km, see Figure 4) that there are no clouds above that contain large ice crystals. On the other hand, large ice particles from below rarely reach these heights, because they are sorted by size (as is the case vice versa with sedimentation) depending on the strength of the updraft: the smaller the ice crystals the higher the altitude (and lower the temperature) they reach. In slow updrafts, that we assume to be present in the cold and thin cirrus, only the smaller liquid-origin ice particles are carried so far up, which is consistent with the observed size range.

We conclude that the low IWCs represent low updrafts and therefore the ice crystal sizes smaller than the in situ  $D_{\rm ice}^{\rm max}$  consists only of in situ-origin cirrus. The ice crystals larger than the in situ  $D_{\rm ice}^{\rm max}$  are almost certainly of liquid-origin. They account for 5% of the data points.

Inspecting now the median (*IWC*: 0.2 - 20 ppmv, Figure 7b) and the thickest (*IWC*: 20 - 2000 ppmv, Figure 7c) cirrus, it is visible that the size ranges are more and more broadening and the concentrations of ice crystals are increasing in comparison with the thinnest cirrus. The overall ice particle size range has grown to maximum sizes of  $\sim$ 500 | 900 µm for the median and thickest cirrus (Figure 7b,c; Table 2), the PSD core size region enclosing 90% of the ice particles expanded to 3-100 | 3-200 µm, while 50% of the ice

particles are found between 12-50 | 10-150  $\mu$ m. The additional IWC is contributed mostly by larger liquid-origin ice particles from below that are more and more present, as large ice crystals commonly dominate IWC.  $N_{\rm ice}$  has grown to a total of 26 | 616  $L^{-1}$  (Table 3). These numbers are indicative for homogeneous in situ ice formation, which is supported by the slightly supersaturated median in-cloud RH<sub>ice</sub> (109 and 112%, see Table 3).

Increasing updrafts are responsible for the expanded ice particle size ranges, increased concentrations of ice particles and increased IWCs (see Figure 7b,c and Table 2), as also discussed by Krämer et al. (2020) based merely on IWC observations. The increasing IWCs stem mainly from increasingly larger ice particles, larger than the in situ  $D_{ice}^{max}$ . As mentioned above, these data points are almost certainly of liquid origin. Their fraction of these data points increase with increasing updraft to 15% and 30%. For ice crystals smaller than  $D_{ice}^{max}$ , the origin cannot be clearly determined, as ice particles of in situ and liquid origin may overlap.

#### **5.2** Middle temperature cirrus (200-220K)






The middle temperature cirrus are shown in the middle row of Figure 7 (d-f). Compared to the coldest cirrus, the amount of available water to grow the ice particles increases to 10-100 ppmv and the in situ  $D_{\rm ice}^{\rm max}$  rises to  $\sim$ 120  $\mu$ m (see vertical cyan line).

The overall ice particle size range has broadened, the maximum sizes of the thinnest | medium | thickest cirrus are  $\sim$ 450 | 800 | 1000 µm (Figure 7d,e,f and Table 2). The respective PSD core size regions enclosing 90% of the ice particles are 3-110 | 3-300 | 3-600 µm, while 50% of the ice particles are found between 25-90 | 20-170 | 20-350 µm. With values of 0.7 | 10.7 | 171 L<sup>-1</sup> (Table 3), the median  $N_{\rm ice}$  again increases as the IWC increases, indicating intensified updrafts. However, compared to the coldest cirrus,  $N_{\rm ice}$  decreases for all three IWC ranges, while the median  $\overline{R}_{\rm ice}$  simultaneously increases to 16 | 19 | 27 µm.

An explanation for the lower  $N_{\rm ice}$  compared to the cold temperatures might be that the homogeneous ice nucleation rates decrease as the temperature increases (Koop et al., 2000; Baumgartner et al., 2022). The broadening of the PSDs in comparison to the cold temperatures (high altitudes), is for the part of the ice particles that form in situ (left of the vertical cyan in situ  $D_{\rm ice}^{\rm max}$  line) a result of the higher amount of water vapor allowing the ice particles to grow to larger sizes. Larger liquid-origin ice particles (right of the vertical cyan in situ  $D_{\rm ice}^{\rm max}$  line) ascending in the updrafts appear at the warmer temperatures (lower altitudes), with their size increasing proportionally to the strength of the updraft. The middle temperature range corresponds to altitudes  $\gtrsim$ 8 km (see Figure 4), thus sedimentation of ice particles from above might also act as an additional source of larger ice particles - however, since at higher altitudes in situ growth of ice particles does not form large particles, this does not seem to be very likely. But, the - in comparison to the coldest temperatures - wider range of the size region enclosing 50% of the data (white contour lines) might be caused by a mixture of sedimented / uplifted liquid origin with in situ origin ice particles. Another

point to note is that the median RH<sub>ice</sub>in the thinnest cirrus is just below 100% (Table 3). This means that sublimation of ice crystals might also influence the PSDs. The ice crystals shrink or dissolve completely in dependence on the degree of subsaturation. It would be interesting to analyze the PSDs with respect to super- and subsaturation, but this is beyond the scope of this study.

# 5.3 The warmest cirrus (220-235K)

The warmest cirrus (bottom row of Figure 7, g-i) have the broadest ice particle size ranges of all temperature intervals. Due to the highest amount of available water to grow the ice particles (100-500 ppmv), the in situ  $D_{\rm ice}^{\rm max}$  is largest (~230 µm). Additionally, at  $\gtrsim 230$  K, in situ-origin ice particles larger than  $D_{\rm ice}^{\rm max}$  may also occur due to aggregation (see Section 2.3.3).

The maximum sizes of the thinnest | medium | thickest cirrus (Figure 7d,e,f) are now  $\sim$ 800 | 1000 | 1000 | 455 µm (Table 2). The respective PSD core size regions enclosing 90% of the ice particles are 30-230 | 15-300 | 3-600 µm, while 50% of the ice particles are found between 100-150 | 30-200 | 3-30/100-300 µm. Medians of  $N_{\rm ice}$  further generally decrease compared to the colder temperatures, but increase with IWC (0.5 | 3.5 | 119 L<sup>-1</sup>, Table 3). With the generally lower  $N_{\rm ice}$ , the median  $\overline{R}_{\rm ice}$  increase, and also increase with IWC (22 | 30 | 39 µm).

The continued broadening of the core PSDs size range (white/black contour lines enclose 50/90% of the data) is due on the one hand to the extended range for in situ ice particle size growth (left of the vertical cyan in situ  $D_{\rm ice}^{\rm max}$  line) and on the other hand because at these warmest temperatures (altitudes  $\gtrsim$ 5km, Figure 4), even larger liquid-origin ice particles (right of the vertical cyan in situ  $D_{\rm ice}^{\rm max}$  line) appear with increasing updrafts. In addition, sedimentation of large ice particles from above likely plays the biggest role at these low altitudes. In the warmest, thickest cirrus with the highest IWCs, this mixture of ice particles from different processes (nucleation, growth, sedimentation) causes the PSD to be nearly flat up to about 300  $\mu$ m and to flatten only at larger sizes.

# 5.4 Cirrus in the $N_i - \overline{R}_i$ space



The properties of the cirrus in the  $N_i - \overline{R}_i{}^3$  space (see Section 3.2) are shown in Figure 8 in the same three different temperature ranges as in Figure 7. From this portrayal of the observations, an impression of the distribution of in situ-origin and liquid-origin cirrus with temperature can be gained in addition to the insights gained from the PSDs. The solid black lines are lines of constant IWC, the colored grid boxes represent occurence frequencies, i.e. this representation combines the most important physical parameters in cirrus. Two thicker lines represent the IWC values of 0.1 and 10 ppmv, whereby cirrus clouds thinner than 0.1 ppmv are considered subvisible, while the 10 ppmv line roughly separates in situ-origin and liquid-

 $<sup>^3</sup>$ as a reminder,  $\overline{R}_{ice}$  is the mean mass ice crystal radius, not to be confused with  $D_{ice}$ (=Dp), which is the actual diameter of the ice crystals.

origin cirrus (Li et al., 2023). Additionally to the classification by Li et al. (2023), we marked an  $N_i - \overline{R}_i$  region at lower IWC and  $N_{\rm ice}$  but larger ice particles that we call 'origin overlap'. Here, with increasingly warmer cirrus clouds, it is not possible to distinguish with certainty what the cirrus origin is, as the in situ-origin ice crystals can become larger and those of liquid-origin can have lower IWCs.

At the *coldest temperatures* (180-200 K), here representing cirrus in the TTL, mainly in situ-origin cirrus with small ice particles occur. Thick liquid-origin cirrus from convection does occur, but are not very common. In the *middle temperature cirrus* (200-220 K), the shift of occurrence frequency pattern to larger sizes nicely shows that the crystals of in situ-origin ice become larger and also that more liquid-origin cirrus appear. The *warmest cirrus* (220-235 K) are dominated by larger ice crystals, where we interpret the most frequently appearing  $N_i - \overline{R}_i$  pairs (white line, enclosing 50% of the obeservations ) as a mixture of larger ice particles sedimented from above (in situ- and liquid-origin) and liquid-origin ice particles uplifted from below.

The core  $N_i - \overline{R}_i$  ranges (white lines) in the temperature intervals are for 180-200 K:  $N_{ice}$  (0.05 - 1 cm<sup>-3</sup>)  $- \overline{R}_{ice}$  (3 - 15  $\mu$ m), for 200-220 K:  $N_{ice}$  (0.05 - 2 cm<sup>-3</sup>)  $- \overline{R}_{ice}$  (5 - 35  $\mu$ m) and for 220-235 K:  $N_{ice}$  (0.02 - 490 2 cm<sup>-3</sup>)  $- \overline{R}_{ice}$  (20 - 55  $\mu$ m).

# 6 Summary and Conclusions



A dataset of airborne measurements of cloud particle size distributions (PSDs) with different cloud spectrometers in is presented here. The entire data set contains 11 airborne field campaigns between 2008 and 2021 with 137.3 hours of measurements ( $\approx 495000$  PSDs) in cirrus clouds in the temperature range of 180 to 235 K and 133.1 hours ( $\approx 480000$  PSDs) spend in mixed-phase and liquid clouds.

The focus of the study is on the analysis of PSDs in cirrus clouds of different microphysical thicknesses for different temperature ranges. For this purpose, first the irregular PSD size grids are synchronized to a logarithmically equidistant grid to allow a combined analysis of the measurements of the different instruments. As next step, frequencies of occurrence of ice crystal concentrations for the different size intervals of the PSDs (PSD heat maps) are portrayed in nine IWC-T tiles <sup>4</sup>. From the specific characteristics of the PSD heatmaps in the IWC-T tiles a more detailed insight into the processes shaping the PSDs under different atmospheric conditions could be derived. The novelty of the findings is the direct link of the microphysical properties of cirrus clouds to the shape of the PSDs and the frequencies of the concentrations of particle sizes.

<sup>&</sup>lt;sup>4</sup>Figure 7 – the cirrus environmental conditions in the nine tiles are: temperature ranges - 180-200 K (coldest cirrus), 200-220 K (middle temperature cirrus) and 220-235 K (warmest cirrus); IWC ranges - 0.001-0.2 ppmv (thin cirrus), 0.2-20 ppmv (median thick cirrus) and 20-2000 ppmv (thick cirrus).

The processes identified to play a role in shaping the PSDs are, on the one hand, the in situ nucleation of small ice crystals (homogeneous or heterogeneous) and their subsequent growth or sublimation caused by small scale temperature fluctuations. On the other hand, uplift and sedimentation of larger in situ- or liquid origin ice crystals influence the PSD shape. These processes overlap and depend on the environmental conditions like temperature, strenghts of updrafts or downdrafts and hence humidity (super- or subsaturated).

Looking in more detail at the processes influencing the PSD shape as a function of cirrus environmental conditions, thickness and temperature, the following picture emerges:

With increasing temperature,






- the maximum size to which in situ formed ice particles can grow rises from  $\sim$ 60 to  $\sim$ 120 to  $\sim$ 230  $\mu$ m at the coldest, middle and warmest temperatures. The corresponding larger ice crystals are of liquid-origin.
- fewer in situ-origin ice particles appear because the homogeneous ice nucleation rate decreases with temperature
  - the cirrus type changes from in situ-origin to liquid-origin (increasing presence of large liquid-origin ice particles, because the updrafts often only lift the air masses to lower altitudes - warmer temperatures).

With increasing IWCs, that correspond to stronger updrafts,

- more small in situ-origin ice particles appear because the homogeneous ice nucleation rate increases with the updraft
- more and larger liquid origin ice particles appear due to stronger uplifts
- the cirrus type changes from in situ-origin to liquid-origin (the lower limit of the liquid-origin fraction of cirrus rises from  $\sim$ 5 to  $\sim$ 15 to  $\sim$ 30 % in thin, median and thick cirrus).

A closer look at the results suggests three characteristic ice crystal size ranges:

- the nucleation / sublimation size interval ( $\sim 3 20 \ \mu m$ ), consisting mostly of freshly nucleated or sublimating in situ-origin ice particles. Except at very cold temperatures, these crystals have a short life time because they grow or sublimate quickly.
  - $\rightarrow$  most frequent ice crystal size range for the *coldest*, thinnest cirrus.
  - the overlap size interval (~20 230 μm), where both in situ-origin and liquid-origin cirrus appear: in situ-origin ice particles by growth or uplift/sedimentation, liquid-origin by uplift/sedimentation or sublimation.
    - → most frequent ice crystal size range of warmer thin cirrus together with medium and thick cirrus,
    - $\rightarrow$  generally the most predominant size range, accounting for about half of all observed ice crystals.
- the uplift / sedimenation size interval (>~230 μm) contains large ice particles which are identified
  to stem from uplifting or sedimenting liquid-origin cirrus; they are the least frequent ice particles,
  however, carry the most ice mass.
  - $\rightarrow$  Thin, medium and thick cirrus are present in this size range.
- In summary, from the ice crystal occurrence patterns in cirrus PSD heat maps the microphysical properties and the associated processes at cold to warm cirrus temperatures in thin to thick cirrus clouds can be derived. Furthermore, these occurrence patterns represent a valuable data set to compare the representation of especially ice clouds in global climate models and in satellite-based remote sensing observations.

## Data availability

The database will be made publicly available when the article is accepted forpublication. Before that, the data can be provided upon request.

## Author contributions

MK designed the study and performed the analyses. NS has generated the synchronized PSD data set and programmed many of the analysis tools. CR performed the simulations. RS, CR and NS contributed to the discussion of the results, improvement of the analysis and preparation of the manuscript.

#### Competing interests


The first author is a member of the editorial board of Atmospheric Chemistry and Physics.

## Acknowledgments

Thanks to a dear colleague who teased me for so long that my new paper would be a Cirrus Guide III until I called it that. Also many thanks to Aaron Bansemer who provided the the SODA software for image processing of the NIXE-CIPgs data.

**Figure 1.** Cirrus PSDs: (a) and (c) mean PSDs, (b) and (d) PSDS along aircraft flights, for two flights during the StratoClim campaign in 2017 out of Nepal. (Dp>3µm is D<sub>ice</sub>).

Figure 2. Cirrus heat maps: (a) and (c) PSDs, (b) and (d)  $N_i - \overline{R}_i$ , for two flights during the StratoClim campaign in 2017 out of Nepal.

**Figure 3.** Cirrus cloud MAID IWC climatology for the scenario with homogeneous and heterogeneous freezing allowed, the concentration of the ice nucleating particles was set to 0.01 cm<sup>-3</sup>, the freezing humidity increases from 110 - 115 % between 235 and 180 K, simulating mineral dust (adapted from Krämer et al., 2016, ).

**Figure 4.** The Jülich In situ Aircraft data set (JULIA) of particle size distributions (PSDs). (a) map of flights (adapted with changes from Bartolomé García et al., 2024), (b) latitude-temperature distribution, (c) altitude-temperature profiles. The data set contains 11 campaigns between 2008 and 2021 (as indicated in the legends) where PSDs were measured; this corresponds to 163 flights, totalling 270.4 hours in cirrus, mixed-phase and liquid clouds).

**Figure 5.** Cloud particle size distributions in 5K temparature intervals from three field campaigns with different aircraft and different pairs of instruments: NIXE-CAPS (StratoClim; CAS-depol+CIPgs), FCDP+2D-S (ATTREX) and CDP+2D-C (CONTRAST; note that the CDP outliers at 50 μm are filtered out); the size of the particles from the optical array probes (CIPgs, 2D-S, 2D-C) is maximum dimension. **Left column:** original irregular size grid of the instruments; **Right column:** synchronized logarithmic equidistant size grid. Note that the transition between each respective pair of instruments is smooth, indicating no systematic errors in the measurements of the individual instruments.

**Figure 6.** Heat maps of cloud particle size distributions (PSDs) in their respective temperature ranges; (a-c): cirrus clouds between 180-235 K ( $\sim$  -90 to -38 $^{\circ}$ C), (d-e): mixed-phase clouds between 235-273 K (-38 to 0 $^{\circ}$ C) and (f): liquid clouds for >273 K (>0 $^{\circ}$ C). Mixed-phase and cirrus clouds are divided into sub-ranges, the coldest temperature range represent the tropical tropopause layer (TTL). Green/black lines: median/mean PSDs; solid black/white contour lines enclose regions containing 90/50% of the data points. The dataset contains 137.2 hours in ice clouds, 69.4 hours in mixed clouds and 63.8 hours in liquid clouds; the measurements hours in the temperature ranges are indicated in each panel.

Figure 7. Cirrus cloud particle size distributions (PSDs) in nine IWC-T (Ice Water Content - Temperature) intervals, color coded by frequencies of occurrence; green/black lines show median/mean PSDs, black/white contour lines enclose 90 / 50% of the data points; the entire data set represent 137.3 hours of measurements. In the respective temperature ranges, the water vapor saturation mixing ratios ( $H_2O_{\rm sat}$ , 26) and the simulated maximum ice crystal sizes for in situorigin cirrus (in situ  $D_{\rm ice}^{\rm max}$ ) are denoted in blue (see also Tables 1 and 2); the T-IWC tile notations are indicated in magenta - the temperature ranges are coldest, middle and warmest, the IWC ranges thin, median and thick.

Figure 8. Frequencies of occurrence of ice particle number  $N_{\rm ice}$  in dependence on size  $\overline{R}_{\rm ice}$  ( $N_i - \overline{R}_i$  heatmaps), in the same three different temperature intervals as in Figure 7; solid black lines: constant IWCs, cirrus below the thicker 0.1 ppmv line are subvisible; the thicker 10 ppmv line roughly separates in situ-origin and liquid-origin cirrus (Li et al., 2023); black/white contour lines enclose 90 / 50% of the data points.

|                                  |            |                          |                                    | Temp       | erature                  | (K)                   |            |                      |                       |
|----------------------------------|------------|--------------------------|------------------------------------|------------|--------------------------|-----------------------|------------|----------------------|-----------------------|
|                                  |            | 190                      |                                    |            | 210                      |                       |            | 230                  |                       |
| $\rm H_2O_{sat,ice} \rightarrow$ | 3 ppmv     |                          |                                    | 35 ppmv    |                          |                       | 300 ppmv   |                      |                       |
| W <sub>(+fluct)</sub><br>(cm/s)  | IWC (ppmv) | $N_{\rm ice}$ $(L^{-1})$ | $\mathbf{D_{ice}^{max}}_{(\mu m)}$ | IWC (ppmv) | $N_{\rm ice}$ $(L^{-1})$ | D <sup>max</sup> (μm) | IWC (ppmv) | $N_{ice}$ $(L^{-1})$ | D <sub>ice</sub> (μm) |
| 50                               | 0.005      | 1 het                    | 6                                  | 0.003      | 1 het                    | 11                    | 5          | 10 het-hom           | 71                    |
| 10                               | 0.003      | 1 het                    | 11                                 | 0.1        | 1 het                    | 56                    | 1          | 1 het                | 141                   |
| 5                                | 0.002      | $1_{\rm het}$            | 18                                 | 0.1        | 1 het                    | 57                    | 40         | 82 het-hom           | 178                   |
| 1                                | 0.002      | 1 het                    | 45                                 | 0.2        | 1 het                    | 113                   | 27         | 25 het-hom           | 179                   |
| 0.1                              | 0.004      | 1 het                    | 56                                 | 0.2        | 0.5 het                  | 89                    | 19         | 15 het-hom           | 224                   |
| Dice upper estimate:             |            | ~60                      | ·                                  | <u> </u>   | ~120                     | ·                     | <u>'</u>   | ~230                 |                       |

**Table 1.** Estimation of maximum  $D_{\rm ice}$  from MAID simulations of cirrus clouds at varying temperatures and updrafts and constant cooling rates superimposed by temperature fluctuations (after Krämer et al., 2016). The updrafts (w) at the point of ice nucleation deviates from the large-scale updraft due to the superimposed small scale fluctuations.

|           | IWC (ppmv)                          |                       |                               |                           |                              |                           |                                |
|-----------|-------------------------------------|-----------------------|-------------------------------|---------------------------|------------------------------|---------------------------|--------------------------------|
|           |                                     | 0.001 -               | 0.2                           | 0.2 - 20                  |                              | 20 - 2000                 |                                |
|           | D <sub>ice</sub> <sup>max,sim</sup> | D <sub>ice</sub> (µm) | D <b>core</b><br>lice<br>(µm) | D <sup>max,obs</sup> (μm) | D <b>core</b><br>ice<br>(µm) | D <sup>max,obs</sup> (μm) | D <b>core</b><br>ice<br>(µm)   |
| 180-200 K | ~60                                 | 350                   | 3 - 70<br>4-15, 30-40         | 500                       | 3 - 100<br>12 - 50           | 900                       | 3 - 200<br>10 - 150            |
| 200-220 K | ~120                                | 450                   | 3 - 110                       | 800                       | 3 <b>-</b> 300               | 1000                      | 3 <b>- 600</b><br>20 - 350     |
| 220-235 K | ~230                                | 800                   | 30 - 230<br>100 - 150         | 1000                      | 15 - 300<br>30 - 200         | 1000                      | 3 <b>- 600</b><br>3-30,100-500 |

 $D_{\mathbf{ice}}^{\mathbf{max,sim}}$ : maximum simulated ice crystal size (in-situ origin);

 $D_{\ensuremath{\mathbf{ice}}}^{\ensuremath{\mathbf{core}}}$  : size range enclosing 90% of the observations;

 $D_{ice}^{\mathbf{max}, \mathbf{obs}}$ : maximum observed ice crystal size (in-situ origin and/or liquid origin).

**Table 2.** Cirrus cloud particle sizes in nine IWC-T (Ice Water Content - Temperature) intervals;  $D_{\rm ice}^{\rm max,sim}$  is the simulated maximum ice crystal sizes for in situ-origin cirrus from Table 1 (= in situ  $D_{\rm ice}^{\rm max}$ ),  $D_{\rm ice}^{\rm max,obs}$  and  $D_{\rm ice}^{\rm core}$  derived from Figure 7.

| Temp (K) | IWC (ppmv)                                               |          |           |  |  |  |
|----------|----------------------------------------------------------|----------|-----------|--|--|--|
|          | 0.001 - 0.2                                              | 0.2 - 20 | 20 - 2000 |  |  |  |
|          | median $\mathbf{N}_{\mathrm{ice}}$ (L <sup>-1</sup> )    |          |           |  |  |  |
| 180-200K | 1.2                                                      | 26       | 616       |  |  |  |
| 200-220K | 0.7                                                      | 10.7     | 171       |  |  |  |
| 220-235K | 0.5                                                      | 3.5      | 119       |  |  |  |
|          | median $\overline{\mathbf{R}}_{\mathrm{ice}}$ ( $\mu$ m) |          |           |  |  |  |
| 180-200K | 7                                                        | 14       | 17        |  |  |  |
| 200-220K | 16                                                       | 19       | 27        |  |  |  |
| 220-235K | 22                                                       | 30       | 39        |  |  |  |
|          | median $\mathbf{RH}_{ice}$ (%)                           |          |           |  |  |  |
| 180-200K | 106                                                      | 109      | 112       |  |  |  |
| 200-220K | 86                                                       | 92       | 101       |  |  |  |
| 220-235K | 83                                                       | 93       | 98        |  |  |  |

**Table 3.** Cirrus median  $N_{\rm ice}$ ,  $\overline{R}_{\rm ice}$  and  $RH_{\rm ice}^{(\star)}$  in nine IWC-T (Ice Water Content - Temperature) intervals, calculated from the combination of all flights shown in Krämer et al. (2020) (their Figure 6) within the respective IWC and temperature intervals. The variation of the 25 and 75%-percentiles around the medians is between -20 and +180% for  $N_{\rm ice}$ , -10 and +40% for  $\overline{R}_{\rm ice}$  and -5 and +20% for  $RH_{\rm ice}$ ; (\*)  $N_{\rm ice}$ ,  $\overline{R}_{\rm ice}$  and  $RH_{\rm ice}$ : ice crystal number concentration, mean mass radius and relative humidity with respect to ice.

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

# **Appendix A: Campaign overviews**

**Figure A.1.** Cloud particle size distributions in 5K temparature intervals for the eleven field campaigns from the JULIA database; the campaigns are grouped by geopgraphical region (note that the measurements during CIRRUS-HL are split into Arctic and mid-latitude); the instruments pairs are noted in the different panels; the size grids are logarithmically equidistantly synchronized

Figure A.2. Heat maps of cloud particle size distributions for the eleven field campaigns from the JULIA database; the campaigns are grouped by geopgraphical region (note that the measurements during CIRRUS-HL are split into Arctic and mid-latitude).

Figure A.3.  $N_i - \overline{R}_i$  heat maps for the eleven field campaigns from the JULIA database; the campaigns are grouped by geopgraphical region (note that the measurements during CIRRUS-HL are split into Arctic and mid-latitude).