# Peer review of "A Microphysics Guide to Cirrus – Part 3: Occurrence patterns of cloud particles"

_EGUsphere, 2025_

## Referee Comment (RC1)

**Referee Comment on "A Microphysics Guide to Cirrus – Part 3: Occurrence patterns of cloud particles"**

**General Comments**

This manuscript represents the third instalment in a well-established series providing a comprehensive microphysical perspective on cirrus clouds. Building upon the foundational work in Cirrus Guides I and II, this paper offers a novel and highly valuable exploration of cirrus particle size distributions (PSDs), leveraging an unprecedentedly large dataset (≈975,000 PSDs over 270 flight hours) from 11 field campaigns. The authors have developed a unique heatmap-based visualization methodology, supported by *in situ* measurements and model simulations, to uncover occurrence patterns of cloud particle sizes under various thermodynamic regimes.

The manuscript is scientifically robust, methodologically sound, and clearly written. It successfully connects microphysical processes with observed PSD features across temperature and IWC regimes, offering insights relevant to both process understanding and climate modelling. Particularly notable is the classification of PSD behaviour into nucleation/sublimation, overlap, and uplift/sedimentation size regimes, and the differentiation between *in situ-* and liquid-origin cirrus using a physically motivated threshold based on simulations.

The integration of measurements, theory, and model results is a major strength. This paper provides not only a valuable resource for the community but also a reference dataset that could serve as a benchmark for global models and remote sensing retrievals. The figures are generally informative, and the inclusion of synchronized PSDs enhances the coherence of multi-instrument data.

However, while the depth of analysis is commendable, some sections (notably in Section 5) could benefit from more concise summarization to maintain focus. Clarifications are also needed regarding the transition between cloud types and the uncertainties involved in origin attribution.

I am also concerned about the lack of the key mechanism of ice crystal aggregation in the model, and would like to be more convinced that comparing Dmax between model and observation is a sound metric for liquid vs in situ origin. The densities, shape factors and growth rate descriptions for ice crystals need a more thorough description for the modelling.

**Specific Comments**

1. **Section 2.3 (Simulations of in situ-origin ice particle sizes)**:
   In the modelling description of MAID you say that you consider diffusional growth, sublimation and sedimentation. But you don't mention aggregation, which can have a very important effect on Dmax, even for in-situ cirrus clouds. Maybe you could comment on this, and how it might affect your findings.

2. **Distinction between in situ- and liquid-origin cirrus**:
   You are using simulated Dmax to discriminate between in situ or liquid formed cirrus clouds. But you are missing aggregation, which may be a key mechanism.

A discussion on the robustness of this attribution in mixed or transitional conditions would be valuable. Is Dmax the best metric? Maybe the shape of the distribution would be more statistically significant? How well does the model reproduce the shape of the distribution? Knowledge of this would allow the reader to assess the methodology.

3. **Figures 7 & 8**:
   These figures are central to the paper's conclusions. At the higher temperature the threshold size for insitu cirrus is 230 micron, but we regularly see that in situ cirrus produce much larger particle sizes, due to aggregation – e.g. see the paper by https://rmets.onlinelibrary.wiley.com/doi/epdf/10.1256/qj.03.138. In addition what growth rates, shape factors and densities were used when modelling diffusional growth – since this is key for calculating Dmax.

4. **Table 3 (Median Nice, Rice, RHice)**:
   This table could benefit from a clearer explanation of how these median values were derived from the climatologies in Cirrus Guide II. Were percentile bounds applied uniformly across all IWC-T tiles?

5. **Cirrus cloud classification terminology**:
   The paper uses terms like "thin," "median thick," and "thick" cirrus. It would be helpful to define these in absolute IWC terms earlier in the paper, perhaps when the IWC-T matrix is first introduced.

6. **Potential for model validation and satellite application**:
   The final paragraph of the conclusions alludes to validation of climate models and remote sensing products. It would strengthen the impact of the paper to expand briefly on how the dataset might be practically used for these purposes (e.g., specific satellite retrieval algorithms or GCM parameterizations).

**Technical Corrections**

- **Start** "Correspnodence" → "Correspondence"
- **Line 66**: "perfom" → "perform"
- **Line 67**: "scienfic" → "scientific"
- **Line 212**: "agreemenent" → "agreement"
- **Line 147**: "liekly" → "likely"
- **Line 551**: "prepartion" → "preparation"
- *In situ* is a Latin phrase meaning "in its original place," and as such, **it should be italicised** especially in scientific contexts.
- "**Database**" is a **compound noun** that has long since been accepted as a single word in both technical and general usage. Similarly "data sets" → "datasets"

---

## Author Comment (AC1)

We thank both referees for taking the time to read the manuscript. Their comments will help us improve it. The reviewer's points are in black, our replies are in blue, and new text in the revised manuscript is in red.

General comment from the authors:

Due to a bug in data processing routines, the $N_{ice}$ and $\overline{R}_{ice}$ data of the ATTREX and POSIDON campaigns were not correct. This affected Figure 8, Table 3 and Figure A3 of the manuscript, which were replaced in the revised manuscript by the corrected versions. However, the statements in the manuscript do not change.

**Responses to Reviewer#2**

The manuscript titled "A Microphysics Guide to Cirrus – Part 3: Occurrence patterns of cloud particles" presents an analysis of cloud particle size distributions (PSDs) derived from 270 hours of measurements, encompassing approximately 975,000 PSDs. The study focuses on cirrus clouds and provides heat maps illustrating PSDs across a range of temperatures and cloud thicknesses. Key findings include the identification of three characteristic ice crystal size ranges:

- Nucleation/Evaporation Size Interval ($\sim$3-20 µm): Predominantly observed in the coldest and thinnest in situ-origin cirrus clouds.

- Overlap Size Interval (20–230 µm): Characterized by the coexistence of both in situ-origin and liquid-origin cirrus crystals.

- Uplift/Sedimentation Size Interval (>230 µm): Consists mainly of liquid-origin ice crystals.

**General comments**

The use of a large data set (270 hours of measurements and 975,000 PSDs) is a significant strength of the study and the volume of data contributes to the robustness and reliability of the findings. The study's focus on cloud particle size distributions (PSDs) in cirrus clouds is highly relevant for advancing the understanding of cloud microphysics, particularly in terms of how PSDs evolve with temperature and cloud thickness.
*Reply* **G1:** Thank you very much for this positive assessment.

The model based identification of the three main ice crystal size intervals is independent from observations which is definitely a weakness of the study.
*Reply* **G2:** We do not quite understand the referee's reasoning here. We have assigned the main ice crystal size intervals from the combined evaluation of simulations and observations, so the identification is not independent of the observations.

Nevertheless, the study provides first limited insights into the processes governing the formation and transformation of cloud particles under different environmental conditions. However, the major weakness of the study is the lack of quantification of in-situ and liquid-origin crystal number and mass fractions, which might have been an objective of the study, but this has not been pursued any further. The effort of investigating quantitatively number and mass fractions of in-situ cirrus and liquid origin cirrus, derived from in situ and liquid origin crystal size distributions would give much more practical use (parametrizations) of these data to the modelling community.

Without this effort the authors cannot claim percentages of in situ and liquid origin crystal number percentages, as is presented in the text.

The additional work will be useful for climate modellers, as the detailed analysis of cirrus cloud microphysics could improve the representation of these clouds in global models. Also it is the third part of a series of papers, thus the authors should insist being more quantitative.

*Reply* **G3:** We would have been pleased to provide a quantification of the number and mass fraction of crystals in situ- and of liquid-origin.

Unfortunately, this is not possible based on the available data. To determine the number and mass fractions of in-situ cirrus and liquid origin cirrus, one would need size-resolved information about the ice particle origin - which we do not have. And I don't see a way to determine the crystal origin across the entire PSD from the measurements. The only reasonably reliable quantification we can make regarding the origin is the fraction of liquid origin ice particles shown in Figure 7:

the blue numbers below the bottom panels are an estimate of the portion of ice particles larger than the maximum size of the crystals formed in situ, that means that they are most certainly of liquid-origin (see also *Reply* **C3-b** and **C4**). Unfortunately, this was not clearly explained in Figure 7 and in the text. We revised Figure 7 and Section 5.1:

' ... The ice crystal larger than the in situ $D_{ice}^{max}$ ice are almost certainly of liquid-origin. ... '

... As mentioned above, these data points are almost certainly of liquid origin. Their fraction of these data points increase with increasing updraft to 15% and 30%. For ice crystals smaller than $D_{ice}^{max}$, the origin cannot be clearly determined, as ice particles of in situ and liquid origin may overlap.

**Criticism**

It would be helpful to provide more detailed descriptions of the measurement methods used during the 270 hours of observation. Specifically, the instruments employed and any potential sources of error should be clearly discussed and quantified.

*Reply* **C1:** Thank you for pointing this out - we described the instruments, their measurement methods, and the uncertainties are described in the first two parts of the study (Krämer et al., 2016; Krämer et al., 2020), information missing there can be found the basic article of Baumgardner et al. (2017). We should have mentioned this, but we think that a repetition would make this manuscript too long.

We added to the first paragraph of Section 3.1:

'The instruments, their measurement methods, and the uncertainties are described in the first two parts of the study (Krämer et al., 2016; Krämer et al., 2020) and in Baumgardner et al. (2017); we therefore refer to these papers for more detailed information.'

1. Line 65: how you differentiate liquid from ice and from mixed phase clouds in the observations?

   *Reply* **C2-1:** We distinguish the clouds according to temperature, T > 273K are liquid clouds, T < 235 K are cirrus ice clouds and in between are the mixed phase clouds (see Figure 6).

2. How do you correct CDP / FCDP data for ice crystals? If not, what's the error and consequences on PSD and subsequent interpretation?
   *Reply* **C2-2:** We did not process the CDP/FCDP measurements ourselves, but received them from the PIs of the measurement respective campaigns. The corresponding publications on the data are listed in Krämer et al. (2016); Krämer et al. (2020).
   We trust the quality of the processing of the data, which has already been peer-reviewed and published in several studies. We also do not believe it is useful or necessary to reiterate the fundamental processing algorithms and instrument uncertainties in each paper, given that they have already been peer-reviewed for specific instruments and campaigns.

3. Have 2DC images been reprocessed for under-sampling in flight direction due to smaller acquisition speed compared to TAS? How is this done? What's the pixel resolution of the 2DC probe? Do you really believe in 60µm crystal concentration data from old 2DC? What would be the uncertainty in size and concentration.
   *Reply* **C2-3:** See answer to point 2, which also applies to 2DC. In addition, as mentioned in the text (Section 3.2, second paragraph; see also Figure 5 and Figure A.1, Appendix A.), the transition between the PSDs of the instrument pairs is always smooth. This indicates that there is no systematic error in the individual measurements (see also *Reply* **S9**).

4. Also 2DS has considerable uncertainties below 100, if not 150µm. You really believe in 25 µm concentration data from those probes? Uncertainties have to be quantified here to judge on interpretation of subsequent results.
   *Reply* **C2-3:** See answer to point 2 and 3, which also applies to 2DS.

Likewise for the model: How realistic is the model as compared to other modelling efforts quantifying in situ versus liquid origin cirrus? What is the estimated error in your model with respect to calculated size thresholds? An answer to this question is necessary to make this study not just a qualitative one.

*Reply* **C3-a:** We only simulated in situ-origin cirrus with the model, liquid-origin cirrus are not included and I am not aware of publications of other such detailed microphysical model studies that include both cloud types.

An error estimate is not made in the model and we do not think that it is necessary considering the method used: the ice crystal maximum sizes are determined for the three temperature intervals from all model runs covering updrafts between 1 to 300 cm/s, INP concentrations between 0.001 and 1 cm$^{-3}$and either low ($\sim$110-120%) or high (130-150%) freezing humidity thresholds (See first paragraph of Section 2.3.2).

From the $D_{ice}^{max}$ of all scenarios of a temperature interval, the largest $D_{ice}^{max}$ value achieved was rounded upwards and taken as the upper size threshold of this temperature range. The span of maximum sizes in the scenarios is quite large for each temperature range: for example, for the coldest cirrus (180-200 K) it is between 6 and 56 µm (Table 1). Here we have defined $D_{ice}^{max}$ to be 60 µm.

We explained the procedure in the manuscript an stated at the end of Section 2.3.3:

'Overall, a conservative estimate from the simulations is that in the three temperature ranges $\sim$190, 210, and 230 K, ice crystals formed in situ do not grow larger than 60, 120, and 230 µm, respectively.'

We think this is sufficient to establish the calculated maximum size thresholds.

As presented, you can't say that below modelled crystal diameter thresholds the crystals are of in situ origin and above of liquid origin. You don't state about overlapping size range.

*Reply* **C3-b:** Our assumption is that larger ice crystals cannot form in situ due to the low water content at low temperatures and are therefore of liquid-origin. Below the threshold, the attribution of the cirrus to the origin is uncertain. We present a clearer explanation in the revised manuscript, see *Reply* **C4**.

Line 407: .... 15% / 30% of the data points are from liquid origin cirrus. That's wrong and authors should better not publish these percentages, the rest is not in situ cirrus percentages. Taking into account the overlapping size range of 20–230m for "co-existing" in situ and liquid origin cirrus, the mentioned fractions may be more like 50 / 75%, or even more? I feel that there is potential to do better. As it stands the study is qualitative. Can you delimit percentages a bit more, taking into account distributions crystals in overlapping in situ and liquid origin size ranges ?

*Reply* **C4:** Thank you for pointing that out, we did not formulate clearly what is meant (see also above, *Reply* **G3 / C3-b**. We revised Figure 7 and write at the end of Section 5.1:

' The increasing IWCs stem mainly from increasingly larger ice particles, larger than the in situ $D_{ice}^{max}$. As mentioned above, these data points are almost certainly of liquid origin. Their fraction of these data points increase with increasing updraft to 15% and 30%. For ice crystals smaller than $D_{ice}^{max}$, the origin cannot be clearly determined, as ice particles of in situ and liquid origin may overlap.'

That would make the study a much better paper, since more quantitative. In addition to corrected number fraction of in situ and liquid origin cirrus, what would be the corresponding mass fractions then? I'd like to insist that authors add an effort to argue percentages in number and mass, with underlying overlapping size distributions of the in situ and liquid cirrus crystal origins. This would make this paper a more valuable paper.

*Reply* **C5:** We agree that it would further improve the study if the number and mass fraction of ice crystals could be derived for in situ- and liquid-origin cirrus. However, as we explained above *Reply* **G3-a**, this is unfortunately not possible.

**Specific comments**

Line 25: Old ICCP report. Update with newer insights
*Reply* **S1:** Updated (IPCC, 2023)

Line 60: Improve differentiation between this study and Bartholomé Garcia et al (2024) publication
*Reply* **S2:** We rephrased the sentence to

'The present study is based on this data set, but focuses on the processes that shape cirrus PSDs, while Bartolomé García et al. (2024) derived PSD parameterizations.'

Line 106: What is the reason to use the mean mass radius/diameter and not the commonly used median mass radius/diameter
*Reply* **S3:** We calculate the mean mass radius for each data point from IWC/Ni (see footnote 1, page 4 of first manuscript version), median mass radii are calculated for data ensembles.'

Line 117: What about in situ cirrus formed of bullet rosette type crystals formed in situ at -35°C?

*Reply* **S4:** The temperature below which ice crystals form instead of liquid cloud droplets is generally considered to be -38°C for the atmosphere, but this is not an absolute value. Therefore, it is not impossible that ice crystals form in situ at -35°C and grow into bullet rosettes. We rephrased the sentence to

'Cirrus of in situ-origin form at temperatures below about -38°C (235 K) heterogeneously or homogeneously on ice nucleating particles containing an insoluble impurity or supercooled solution aerosol particles (Luebke et al., 2016; Krämer et al., 2016).'

Line 127: a new nucleation event... I don't think that you can distinguish between in situ cirrus and liquid origin cirrus events. There might be rather a continuum with dominating in situ or liquid origin clouds in numerous cases of cirrus cloud formation.

*Reply* **S5:** Based on the PSDs, one can get an idea of whether the cirrus is of in situ or of liquid origin and whether a new nucleation event has occurred. This is shown in Figure 1, panels b and d, and discussed in this section (2.2).

Line 149: can you prove the argument with gravity waves? Otherwise skip...

*Reply* **S6:** We replaced 'gravity waves' by 'strong convective conditions', as this is more reliable information.

Line 158: Please summarize numbers in Table 1 in few clear sentences!

*Reply* **S7:** We rephrased the sentence to

'To get an impression on its dependence on temperature: the amount of water vapor at water saturation, $H2O_{sat,ice}$, is 3, 35, 300 ppmv at 190, 210, 230 K (see Table 1).'

Line 274: Mention mass diameter (which) relationship used here.

*Reply* **S8:** Done.

Line 2901-290: This sentence has to be written more carefully. We know about significant errors in PSD retrievals from OAP probes. Also you may hide factors of up to 5 in concentration differences between probe pairs in the transition size range on a log scale.

*Reply* **S9**: We rephrased the sentence to

'Finally, looking at the transitions between the PSDs of the instrument pairs reveals no discrepancies so large that the measurements would have to be discarded. The transitions falls within the known instrumental uncertainties, which unfortunately are generally not insignificant for cloud spectrometers.'

Line 329: It seems that you mis up aerosol spectra from optical spectrometer with cloud spectra, since the largest aerosol particles should be activated, no? What's the consequence of that?

*Reply* **S10:** The measurements were made with the cloud probes, but in cloud free air, as now stated more clearly in the revised manuscript:

'However, these are most likely not cloud droplets but aerosol particles in cloud free air, as the warm temperatures are found near the ground, ... '

Line 327: Those larger droplets may produce secondary ice, thus your argument is not supportable...

*Reply* **S11:** I don't understand what is meant here - larger drops produce secondary ice?

Line 331: Why the Dollner (2024) algorithm is not applied then?

*Reply* **S12:** We did not distinguish between aerosol and cloud events because our main focus is on cirrus clouds, at altitudes where aerosol particles rarely interfere with the ice clouds. The Dollner (2024) algorithm was only developed after our analyses had already been completed - repeating the analyses was deemed to be not needed for our purposes.

Line 345-348: Unclear sentence. Please clarify

*Reply* **S13:** We rephrased the sentence to

'The yellow-red color code indicate the frequencies of the ice particles across the entire size range.'

Line 474-475: unreadable!

*Reply* **S14:** We rephrased the sentence to

'The core $N_i - \overline{R}_i$ ranges (white lines) in the temperature intervals are for 180-200 K: $N_{ice}$ (0.05 - 1 cm$^{-3}$) $- \overline{R}_{ice}$ ( 3 - 15 µm), for 200-220 K: $N_{ice}$ (0.05 - 2 cm$^{-3}$) $- \overline{R}_{ice}$ ( 5 - 35 µm) and for 220-235 K: $N_{ice}$ (0.02 - 2 cm$^{-3}$) $- \overline{R}_{ice}$ (20 - 55 µm).'

Figure 3: what is s09 what is MD, figure caption simply uncomprehensible

*Reply* **S15:** Good comment - this was a work title and should have been worded even better, thanks for reading so carefully! We rephrased the caption to

'Cirrus cloud MAID IWC climatology for the scenario with homogeneous and heterogeneous freezing allowed, the concentration of the ice nucleating particles was set to 0.01 cm$^{-3}$, the freezing humidity increases from 110 - 115 % between 235 and 180 K, simulating mineral dust (adapted from Krämer et al., 2016, ).'

Figure 4a, then 4b,c : chose same colours for identical campaigns

*Reply* **S16:** Done.

**Technical comments**

Line 53 improve wording: ... in dependence..

*Reply* **T1:** We rephrased the sentence to

'The study focuses on characterizing the shapes of ice particles in relation to their size under various atmospheric conditions.'

Line 83-84: sentence unclear

*Reply* **T2:** We rephrased the sentence to

'Whether some of the physical processes that form the PSDs can be identified from the measurements depends to a certain degree on how these are displayed.'

Line 107 discussion Fig 2: black and white contour lines almost invisible

*Reply* **T3:** Figure 2 has been modified.

Line 124: wording: ....on a time scale faster with....?

*Reply* **T4:** The sentence has been modified to

'... these particles grow or shrink by diffusional growth/sublimation at a faster rate as the cooling or warming rate increases ...'

Line 126: wording: ....sediment out...
*Reply* **T5:** The wording has been modified to

' ... seddle out ... '

Line 128: wording: ....shrink back...
*Reply* **T6:** We could not find any improvement in the wording without changing the meaning.

Line 157: wording: For an impression....
*Reply* **T7:** The wording has been modified to

'To get an impression ... '

Line 240: replace should by is....
*Reply* **T8:** I can't find 'should by is' ...

Line 242: replace 'secondly'...
*Reply* **T9:** The wording has been modified to

'every second' ...

Line 373/ contour lines really hard to see
*Reply* **T10:** 'Yes, the plots are visually busy, but they contain a lot of information. Increasing the thickness of the contour lines would make the median and mean lines harder to distinguish. However, we think that with variable display options, readers can zoom in to explore the plots in greater detail and more easily identify specific features.'

**References**

Bartolomé García, I., Sourdeval, O., Spang, R., and Krämer, M.: Technical note: Bimodal parameterizations of in situ ice cloud particle size distributions, Atmospheric Chemistry and Physics, 24, 1699–1716, https://doi.org/10.5194/acp-24-1699-2024, 2024.

Baumgardner, D., Abel, S. J., Axisa, D., Cotton, R., Crosier, J., Field, P., Gurganus, C., Heymsfield, A., Korolev, A., Krämer, M., Lawson, P., McFarquhar, G., Ulanowski, Z., and Um, J.: Cloud Ice Properties: In Situ Measurement Challenges; Chapter 9 of 'Ice Formation and Evolution in Clouds and Precipitation: Measurement and Modeling Challenges', Meteorol. Monographs, https://doi.org/10.1175/AMSMONOGRAPHS-D-16-0011.1, 2017.

Bunz, H., Benz, S., Gensch, I., and Krämer, M.: MAID: a model to simulate UT/LS aerosols and ice clouds, Envir. Res. Lett., 3, 035 001 (8pp), https://doi.org/10.1088/1748-9326/3/3/035001, 2008.

Gallagher, M. W., Connolly, P. J., Whiteway, J., Figueras-Nieto, D., Flynn, M., Choularton, T. W., Bower, K. N., Cook, C., Busen, R., and Hacker, J.: An overview of the microphysical structure of cirrus clouds observed during EMERALD-1, Quarterly Journal of the Royal Meteorological Society, 131, 1143–1169, https://doi.org/https://doi.org/10.1256/qj.03.138, 2005.

IPCC: Climate Change 2021 – The Physical Science Basis: Working Group I Contribution to the Sixth Assessment Report of the Intergovernmental Panel on Climate Change, Cambridge University Press, 2023.

Krämer, M., Rolf, C., Spelten, N., Afchine, A., Fahey, D., Jensen, E., Khaykin, S., Kuhn, T., Lawson, P., Lykov, A., Pan, L. L., Riese, M., Rollins, A., Stroh, F., Thornberry, T., Wolf, V., Woods, S., Spichtinger, P., Quaas, J., and Sourdeval, O.: A microphysics guide to cirrus – Part 2: Climatologies of clouds and humidity from observations, Atmospheric Chemistry and Physics (highlight article), 20, 12 569–12 608, https://doi.org/10.5194/acp-20-12569-2020, 2020.

Krämer, M., Rolf, C., Luebke, A., Afchine, A., Spelten, N., Costa, A., Meyer, J., Zoeger, M., Smith, J., Herman, R. L., Buchholz, B., Ebert, V., Baumgardner, D., Borrmann, S., Klingebiel, M., and Avallone, L.: A microphysics guide to cirrus clouds - Part 1: Cirrus types, Atmospheric Chemistry and Physics, 16, 3463–3483, https://doi.org/{10.5194/acp-16-3463-2016}, 2016.

Luebke, A. E., Afchine, A., Costa, A., Grooss, J.-U., Meyer, J., Rolf, C., Spelten, N., Avallone, L. M., Baumgardner, D., and Krämer, M.: The origin of midlatitude ice clouds and the resulting influence on their microphysical properties, Atmospheric Chemistry and Physics, 16, 5793–5809, https://doi.org/{10.5194/acp-16-5793-2016}, 2016.

Spichtinger, P.: Aggregation of ice crystals in a cirrus cloud model, by Kienast-Sjögren, E., Spichtinger, P. and K. Gierens, EGU 2011, personal information, 2023.

Sölch, I. and Kärcher, B.: A large-eddy model for cirrus clouds with explicit aerosol and ice microphysics and Lagrangian ice particle tracking, Quarterly Journal of the Royal Meteorological Society, 136, 2074–2093, https://doi.org/https://doi.org/10.1002/qj.689, 2010.

Wernli, H., Boettcher, M., Joos, H., Miltenberger, A. K., and Spichtinger, P.: A trajectory-based classification of ERA-Interim ice clouds in the region of the North Atlantic storm track, Geophysical Research Letters, 43, 6657–6664, https://doi.org/10.1002/2016GL068922, 2016.

Wolf, V., Kuhn, T., Milz, M., Voelger, P., Krämer, M., and Rolf, C.: Arctic ice clouds over northern Sweden: microphysical properties studied with the Balloon-borne Ice Cloud particle Imager B-ICI, Atmospheric Chemistry and Physics, 18, 17 371–17 386, https://doi.org/10.5194/acp-18-17371-2018, 2018.

---

## Author Comment (AC2)

We thank both referees for taking the time to read the manuscript. Their comments will help us improve it. The reviewer's points are in black, our replies are in blue, and new text in the revised manuscript is in red.

General comment from the authors:

Due to a bug in data processing routines, the $N_{ice}$ and $\overline{R}_{ice}$ data of the ATTREX and POSIDON campaigns were not correct. This affected Figure 8, Table 3 and Figure A3 of the manuscript, which were replaced in the revised manuscript by the corrected versions. However, the statements in the manuscript do not change.

**Responses to Reviewer#1**

**General Comments**

This manuscript represents the third instalment in a well-established series providing a comprehensive microphysical perspective on cirrus clouds. Building upon the foundational work in Cirrus Guides I and II, this paper o@ers a novel and highly valuable exploration of cirrus particle size distributions (PSDs), leveraging an unprecedentedly large data set ($\sim$975,000 PSDs over 270 flight hours) from 11 field campaigns. The authors have developed a unique heat map-based visualization methodology, supported by in situ measurements and model simulations, to uncover occurrence patterns of cloud particle sizes under various thermodynamic regimes.

The manuscript is scientifically robust, methodologically sound, and clearly written. It successfully connects microphysical processes with observed PSD features across temperature and IWC regimes, offering insights relevant to both process understanding and climate modeling. Particularly notable is the classification of PSD behavior into nucleation/sublimation, overlap, and uplift/sedimentation size regimes, and the differentiation between in situ- and liquid-origin cirrus using a physically motivated threshold based on simulations.

The integration of measurements, theory, and model results is a major strength. This paper provides not only a valuable resource for the community but also a reference data set that could serve as a benchmark for global models and remote sensing retrievals. The figures are generally informative, and the inclusion of synchronized PSDs enhances the coherence of multi-instrument data.

*Reply* **G1:** Thank you for the positive evaluation of the manuscript.

However, while the depth of analysis is commendable, some sections (notably in Section 5) could benefit from more concise summarization to maintain focus. Clarifications are also needed regarding the transition between cloud types and the uncertainties involved in origin attribution.

*Reply* **G2:** See *Reply* **S1**.

I am also concerned about the lack of the key mechanism of ice crystal aggregation in the model, and would like to be more convinced that comparing Dmax between model and observation is a sound metric for liquid vs in situ origin. The densities, shape factors and growth rate descriptions for ice crystals need a more thorough description for the modelling.

*Reply* **G3:** See *Reply* **S1** and **S3-b**.

Specific Comments

1. Section 2.3 (Simulations of in situ-origin ice particle sizes):
   In the modelling description of MAID you say that you consider diffusional growth, sublimation and sedimentation. But you don't mention aggregation, which can have a very important effect on Dmax, even for in-situ cirrus clouds. Maybe you could comment on this, and how it might affect your findings.

   *Reply* **S1:** We mentioned aggregation in a paragraph at the **end of Section 2.3.3**:

   *'Aggregation of the in situ-origin ice crystals could produce larger crystals, but this process plays a -minor- role only at the warmest cirrus temperatures (Spichtinger, 2023; Sölch and Kärcher, 2010) and a large role for ice crystals falling from this altitudes into warmer saturated or supersaturated regions, that we do not consider in this study.'*

   We extended this paragraph, and the new text reads as follows:

   'Aggregation of the in situ-origin ice crystals could produce larger crystals, but this process plays a -minor- role only at the warmest cirrus temperatures (Spichtinger, 2023; Sölch and Kärcher, 2010), because aggregation has a strong dependence on the ice particle size and weakens with ice mass, number concentration and temperature. Therefore, aggregation occurs mainly in fall streaks, i.e. at temperatures that are usually higher than those considered here, where the ice particles are large. This can also be seen in Gallagher et al. (2005), where aggregation is observed in fall streaks at temperatures between about 230 and 240 K, and is also discussed in a modelling study by Sölch and Kärcher (2010).

   Further, the maximum ice crystal sizes from the MAID simulations are supported by observations of Wolf et al. (2018), which are now mentioned in the revised manuscript:

   'The maximum ice crystal sizes of ice from the MAID simulations are supported by observations of Wolf et al. (2018), who found an average maximum size of $\sim$ 140 µm in their measurements in in situ-origin cirrus.'

   However, we agree that we cannot completely rule out the possibility that aggregation might occur in the warmest temperature interval (220-235 K), in particular at high IWCs. In such cases, the limiting size for in-situ ice particles might be larger. We changed the manuscript accordingly at the end of Section 2.3.3:

   'Overall, a conservative estimate from the simulations is that in the three temperature ranges $\sim$190, 210, and 230 K, ice crystals formed in situ do not grow larger than 60, 120, and 230 µm, respectively. However, we cannot rule out that at $> 230$ K , in situ-origin ice particles larger than $D_{ice}^{max}$ may also occur, resulting from aggregation.'

   and at the beginning of Section 5.3:

   'Due to the highest amount of available water to grow the ice particles (100-500 ppmv), the in situ $D_{ice}^{max}$ is largest ($\sim$230 µm). Additionally, at temperatures $> 230$ K, in situ-origin ice particles larger than $D_{ice}^{max}$ may also occur due to aggregation (see Section 5.3.')

2. Distinction between in situ- and liquid-origin cirrus:

You are using simulated Dmax to discriminate between in situ or liquid formed cirrus clouds. But you are missing aggregation, which may be a key mechanism. A discussion on the robustness of this attribution in mixed or transitional conditions would be valuable.

*Reply* **S2-a:** See response to point 1.

Is Dmax the best metric? Maybe the shape of the distribution would be more statistically significant? How well does the model reproduce the shape of the distribution? Knowledge of this would allow the reader to assess the methodology.

*Reply* **S2-b:** That's a good point—we also thought first we could use the shape of the size distribution. The idea was that, on average, in-situ origin cirrus have a unimodal size distribution, while liquid origins have two modes. However, that turned out not to be true. In-situ origin cirrus often also have two modes, since heterogeneous freezing occurs first, followed by homogeneous freezing.

From the observations it became clear that size is the most valid distinguishing feature, as can be seen from Figure 1d, bottom panel:

[Figure]

The best possible way to determine the origin of cirrus is to calculate air mass backward trajectories, as done by Wernli et al. (2016) and Luebke et al. (2016). However, this is very time-consuming for such a large data set and also not free from uncertainties.

We believe that sorting by size, although imperfect, is a robust method to determine the origin of cirrus clouds.

We did not check whether the simulated distributions matches the measured, since we did not know the history of the observed PSDs. However, Figure 1 shows an example of the time evolution of a cirrus PSD in the temperature range <190K (left panel) and the heat map of the observations in the same temperature range (Figure 7a) and low IWC, to exclude liquid origin cirrus as good as possible.

The simulated PSD has the peak size at a bit larger sizes and lower concentrations than median of the heat map, because of the very slow updraft and low INP number (the PSDs are from purely heterogeneous freezing). Nevertheless, this comparison shows that the PSDs are well represented by the model.

[Figure]

Figure 1: left: MAID simulation, right: observations (Figure 7a)

3. Figures 7 & 8:
   These figures are central to the paper's conclusions. At the higher temperature the threshold size for in situ cirrus is 230 micron, but we regularly see that in situ cirrus produce much larger particle sizes, due to aggregation – e.g. see the paper by `https://rmets.onlinelibrary.wiley.com/doi/epdf/10.1256/qj.03.138`.

   *Reply* **S3-a:** As you rightly point out, the large ice crystals formed by aggregation are usually found at warmer temperatures in fall streaks, often above the temperatures we are considering here (see also *Reply* **S1** ). So our largest IWC-T interval (Figure 7i and Figure 8c) may also contain ice crystals formed by aggregation. This is mentioned now at the beginning of Section 5.3:

   'Due to the highest amount of available water to grow the ice particles (100-500 ppmv), the in situ $D_{ice}^{max}$ is largest ($\sim$230 µm). Additionally, at temperatures > 230 K, in situ-origin ice particles larger than $D_{ice}^{max}$ may also occur due to aggregation (see Section 5.3.')

   In addition what growth rates, shape factors and densities were used when modelling diffusional growth – since this is key for calculating Dmax.

   *Reply* **S3-b:** The ice particle growth is calculated from the difference between the partial pressure of the corresponding trace gas in the bulk and its vapor pressure at the particle surface (see Bunz et al., 2008). The particles are assumed to be spherical[⋆] with an accommodation coefficient of 1 for water molecules and the density of ice crystals is set to 0.91 g/cm$^3$.
   [⋆]: This assumption can be made for the size range of in situ-origin cirrus clouds. Very different shapes are only found in liquid-origin cirrus clouds, as these evolved at warmer temperatures and with a higher water content in the air (see Wolf et al., 2018).

4. Table 3 (Median Nice, Rice, RHice):
   This table could benefit from a clearer explanation of how these median values were derived from the climatologies in Cirrus Guide II. Were percentile bounds applied uniformly across all IWC-T tiles?

*Reply* **S4:** We have changed the corresponding text in the caption of Table 3 to
'Cirrus median $N_{ice}$, $\overline{R}_{ice}$ and $RH_{ice}{}^{(\star)}$ in nine IWC-T (Ice Water Content - Temperature) intervals, calculated from the combination of all flights shown in Krämer et al. (2020) (their Figure 6) within the respective IWC and temperature intervals. ...'
And yes, the percentile limits were applied uniformly to all IWC-T tiles.

5. Cirrus cloud classification terminology:
   The paper uses terms like "thin," "median thick," and "thick" cirrus. It would be helpful to define these in absolute IWC terms earlier in the paper, perhaps when the IWC-T matrix is first introduced.

   *Reply* **S5:** We introduce the names for the temperature and IWC ranges at the beginning of Section 5, where the IWC-T matrix appears for the first time. To make them easier to find at any time, we have now defined them in Figure 7 and the corresponding caption. We hope this makes things clearer.

6. Potential for model validation and satellite application:
   The final paragraph of the conclusions alludes to validation of climate models and remote sensing products. It would strengthen the impact of the paper to expand briefly on how the data set might be practically used for these purposes (e.g., specific satellite retrieval algorithms or GCM parameterizations).

   *Reply* **S-6:** You are right that such a statement for specific models and satellite retrieval would strengthen the impact of the paper, but since this is not so easy to say and might be different for differing models / algorithms, we have formulated this sentence in a somewhat more general way:

   '... Furthermore, these occurrence patterns represent a valuable data set to compare the representation of especially ice clouds in global climate models and in satellite-based remote sensing observations.'

**Technical Corrections**

- Start "Correspnodence" → "Correspondence"
  *Reply* **T-1:** Done.

- Line 66: "perfom" → "perform"
  *Reply* **T-2:** Done.

- Line 67: "scienfic" → "scientific"
  *Reply* **T-3** Done.

- Line 212: "agreemenent" → "agreement"
  *Reply* **T-4:** Done.

- Line 147: "liekly" → "likely"
  *Reply* **T-5:** Done.

- Line 551: "prepartion" → "preparation"
  *Reply* **T-6:** Done.

- In situ is a Latin phrase meaning "in its original place," and as such, it should be italicised especially in scientific contexts.
  *Reply* **T-7:** I had a long discussion with Copernicus about the notation of in situ origin (and liquid origin), and the one used here is what we agreed on in Krämer et al. (2016). Therefore, I think it's consistent to keep it this way.

- "Database" is a compound noun that has long since been accepted as a single word in both technical and general usage.
  *Reply* **T-8:** Changed.

**References**

Bartolomé García, I., Sourdeval, O., Spang, R., and Krämer, M.: Technical note: Bimodal parameterizations of in situ ice cloud particle size distributions, Atmospheric Chemistry and Physics, 24, 1699–1716, https://doi.org/10.5194/acp-24-1699-2024, 2024.

Baumgardner, D., Abel, S. J., Axisa, D., Cotton, R., Crosier, J., Field, P., Gurganus, C., Heymsfield, A., Korolev, A., Krämer, M., Lawson, P., McFarquhar, G., Ulanowski, Z., and Um, J.: Cloud Ice Properties: In Situ Measurement Challenges; Chapter 9 of 'Ice Formation and Evolution in Clouds and Precipitation: Measurement and Modeling Challenges', Meteorol. Monographs, https://doi.org/10.1175/AMSMONOGRAPHS-D-16-0011.1, 2017.

Bunz, H., Benz, S., Gensch, I., and Krämer, M.: MAID: a model to simulate UT/LS aerosols and ice clouds, Envir. Res. Lett., 3, 035 001 (8pp), https://doi.org/10.1088/1748-9326/3/3/035001, 2008.

Gallagher, M. W., Connolly, P. J., Whiteway, J., Figueras-Nieto, D., Flynn, M., Choularton, T. W., Bower, K. N., Cook, C., Busen, R., and Hacker, J.: An overview of the microphysical structure of cirrus clouds observed during EMERALD-1, Quarterly Journal of the Royal Meteorological Society, 131, 1143–1169, https://doi.org/https://doi.org/10.1256/qj.03.138, 2005.

IPCC: Climate Change 2021 – The Physical Science Basis: Working Group I Contribution to the Sixth Assessment Report of the Intergovernmental Panel on Climate Change, Cambridge University Press, 2023.

Krämer, M., Rolf, C., Spelten, N., Afchine, A., Fahey, D., Jensen, E., Khaykin, S., Kuhn, T., Lawson, P., Lykov, A., Pan, L. L., Riese, M., Rollins, A., Stroh, F., Thornberry, T., Wolf, V., Woods, S., Spichtinger, P., Quaas, J., and Sourdeval, O.: A microphysics guide to cirrus – Part 2: Climatologies of clouds and humidity from observations, Atmospheric Chemistry and Physics (highlight article), 20, 12 569–12 608, https://doi.org/10.5194/acp-20-12569-2020, 2020.

Krämer, M., Rolf, C., Luebke, A., Afchine, A., Spelten, N., Costa, A., Meyer, J., Zoeger, M., Smith, J., Herman, R. L., Buchholz, B., Ebert, V., Baumgardner, D., Borrmann, S., Klingebiel, M., and Avallone, L.: A microphysics guide to cirrus clouds - Part 1: Cirrus types, Atmospheric Chemistry and Physics, 16, 3463–3483, https://doi.org/{10.5194/acp-16-3463-2016}, 2016.

Luebke, A. E., Afchine, A., Costa, A., Grooss, J.-U., Meyer, J., Rolf, C., Spelten, N., Avallone, L. M., Baumgardner, D., and Krämer, M.: The origin of midlatitude ice clouds and the resulting influence on their microphysical properties, Atmospheric Chemistry and Physics, 16, 5793–5809, https://doi.org/{10.5194/acp-16-5793-2016}, 2016.

Spichtinger, P.: Aggregation of ice crystals in a cirrus cloud model, by Kienast-Sjögren, E., Spichtinger, P. and K. Gierens, EGU 2011, personal information, 2023.

Sölch, I. and Kärcher, B.: A large-eddy model for cirrus clouds with explicit aerosol and ice microphysics and Lagrangian ice particle tracking, Quarterly Journal of the Royal Meteorological Society, 136, 2074–2093, https://doi.org/https://doi.org/10.1002/qj.689, 2010.

Wernli, H., Boettcher, M., Joos, H., Miltenberger, A. K., and Spichtinger, P.: A trajectory-based classification of ERA-Interim ice clouds in the region of the North Atlantic storm track, Geophysical Research Letters, 43, 6657–6664, https://doi.org/10.1002/2016GL068922, 2016.

Wolf, V., Kuhn, T., Milz, M., Voelger, P., Krämer, M., and Rolf, C.: Arctic ice clouds over northern Sweden: microphysical properties studied with the Balloon-borne Ice Cloud particle Imager B-ICI, Atmospheric Chemistry and Physics, 18, 17 371–17 386, https://doi.org/10.5194/acp-18-17371-2018, 2018.